# Pathological Alterations in Heart Mitochondria in a Rat Model of Isoprenaline-Induced Myocardial Injury and Their Correction with Water-Soluble Taxifolin

**DOI:** 10.3390/ijms252111596

**Published:** 2024-10-29

**Authors:** Natalia V. Belosludtseva, Tatyana A. Uryupina, Lyubov L. Pavlik, Irina B. Mikheeva, Eugeny Yu. Talanov, Natalya I. Venediktova, Dmitriy A. Serov, Mikhail R. Stepanov, Mikhail A. Ananyan, Galina D. Mironova

**Affiliations:** 1Institute of Theoretical and Experimental Biophysics, Russian Academy of Sciences, Institutskaya 3, 142290 Pushchino, Russia; 2Prokhorov General Physics Institute of the Russian Academy of Sciences, Vavilov St. 38, 119991 Moscow, Russia; 3Advanced Technologies Ltd., 119333 Moscow, Russia

**Keywords:** ISO-induced myocardial injury, water-soluble taxifolin, mitochondria, ultrastructural alterations, oxidation phosphorylation, lipid peroxidation, antioxidant enzymes, oxidative damage

## Abstract

Mitochondrial damage and associated oxidative stress are considered to be major contributory factors in cardiac pathology. One of the most potent naturally occurring antioxidants is taxifolin, especially in its water-soluble form. Herein, the effect of a 14-day course of the peroral application of the water-soluble taxifolin (aqTAX, 15 mg/kg of body weight) on the progression of ultrastructural and functional disorders in mitochondria and the heart’s electrical activity in a rat model of myocardial injury induced with isoprenaline (ISO, 150 mg/kg/day for two consecutive days, *subcut*) was studied. The delayed ISO-induced myocardial damage was accompanied by an increase in the duration of RR and QT intervals, and long-term application of aqTAX partially restored the disturbed intraventricular conduction. It was shown that the injections of ISO lead to profound ultrastructural alterations of myofibrils and mitochondria in cardiomyocytes in the left ventricle myocardium, including the impairment of the ordered arrangement of mitochondria between myofibrils as well as a decrease in the size and the number of these organelles per unit area. In addition, a reduction in the protein level of the subunits of the respiratory chain complexes I-V and the activity of the antioxidant enzymes catalase, glutathione peroxidase, and Mn-SOD in mitochondria was observed. The application of aqTAX caused an increase in the efficiency of oxidation phosphorylation and a partial restoration of the morphometric parameters of mitochondria in the heart tissue of animals with the experimental pathology. These beneficial effects of aqTAX are associated with the inhibition of lipid peroxidation and the normalization of the enzymatic activities of glutathione peroxidase and Mn-SOD in rat cardiac mitochondria, which may reduce the oxidative damage to the organelles. Taken together, these data allow one to consider this compound as a promising cardioprotector in the complex therapy of heart failure.

## 1. Introduction

According to the annual World Health Statistics reports, cardiovascular diseases (CVDs) remain the main cause of mortality and a serious pathological problem worldwide. Despite some progress made in the treatment of cardiac pathologies, the mortality from this group of diseases grows steadily every year [1]. Thus, according to the reports, about 17.9 million patients with heart diseases died in 2023, which amounts to 35% of all deaths in the world. The most severe myocardial pathologies are ischemic heart disease, hypertension, cardiomyopathies, cardiac hypertrophy, and chronic cardiac insufficiency. In turn, cardiac hypertrophy is considered to be the main risk factor for many CVDs [2,3].

The activity of the sympathetic nerve system is important for the cardiovascular homeostasis, but its hyperactivity under stressful conditions is linked to the progression of CVD, including oxidative stress and energy deficits, which results in myocardial remodeling, infarction, and cardiac failure [4,5,6]. It is known that excessive stimulation of β-adrenergic receptors, observed under severe stress or exposure to agonists, in particular, isoprenaline (ISO), contributes to heart hypertrophy, which may lead to sudden cardiac death at remote terms [7,8]. It was also found that prolonged stimulation of these receptors by ISO causes excessive production of reactive oxygen species (ROS) and cell death, resulting in myocardial infarction [9,10,11].

Functional and ultrastructural abnormalities in cardiac mitochondria are among the main pathophysiological mechanisms underlying myocardial ischemia–reperfusion injury [12]. In this case, cardiomyocytes change over to predominantly consuming glucose through glycolysis, which is accompanied by a decrease in the activity of the enzymes of the tricarboxylic acids cycle, electron transport chain complexes, and the intensity of β-oxidation in mitochondria. The disturbance of the ATP synthesis observed during the mitochondrial dysfunction leads to energy deficit in the tissue and a decline in the heart muscle function [13,14]. Along with compromised energy metabolism, the dysfunction of mitochondria may result in the destruction of redox homeostasis in cells and an increase in ROS accumulation, the main source of which in pathologies is the electron transport chain. This may be associated with changes in the activity of major mitochondrial antioxidant enzymes, such as catalase, manganese superoxide dismutase (Mn-SOD), and glutathione peroxidase [15,16,17].

Since mitochondria constitute 25–30% of the myocardium weight, their dysfunction and excessive generation of ROS significantly contribute to the progression of CVD and attendant oxidative stress [15,18,19]. In turn, oxidative stress is the disturbance of the redox homeostasis, as a result of which the level of ROS production significantly transcends the capacity of the antioxidant enzymes of the body to utilize them. A gradual decline in the activity of these enzymes with the retention of enhanced ROS formation may lead to chronic oxidative myocardial damage [18,19].

Studies showed that oxidative-induced damage to the myocardium can be eliminated by some natural polyphenols [20,21]. Dihydroquercetin, also known as taxifolin (TAX), is a natural flavonoid possessing potent antioxidant and anti-inflammatory properties. It widely occurs in various plant sources, including conifers such as Siberian larch. It was found that TAX has a more powerful antioxidant effect and is retained in the blood for a longer time after administration to animals, compared with the familiar analogue quercetin [22,23]. The main characteristic feature of TAX is the presence of saturated hydrocarbon bonds in its pyran fragment, which distinguishes it from other flavonoids such as quercetin. Due to this unique structure, the compound is capable of binding to free radicals, thereby protecting cells against oxidative stress, and affect interactions with other molecular targets in the body. Recently, we have found that the water-soluble form aqua taxifolin (aqTAX) (as a complex of TAX with polyvinylpyrrolidone (PVP)) exhibits pronounced antihypoxic and cytoprotective activities in cerebral cortex cell cultures during oxygen-glucose deprivation (OGD) at much lower concentrations compared with the classical (powder) form of TAX [24]. The higher bioavailability of aqTAX appears to be due to electrostatic interactions with PVP, as evidenced via nuclear magnetic resonance spectroscopy. As a result, in an ischemic medium, aqTAX suppresses the OGD-induced increase in the concentration of Ca^2+^ in the cytosol of neurons and astrocytes, almost entirely eliminating necrotic cell death, which does not happen when using classical TAX [24].

The effect of aqTAX on the mechanisms of mitochondrial dysfunction is poorly understood. At the same time, effective therapy aimed at decreasing the oxidative damage and mitochondrial dysfunction can be considered as a qualitatively novel strategy for suppressing high sensitivity to prohypertrophic stimuli and ameliorating heart failure [20,21].

In this study, we investigated the effect of a 14-day course of peroral administration of aqTAX (15 mg/kg of body weight) to animals on the electrophysiological characteristics of the myocardium and structural, biochemical, and functional indices of mitochondria of rats with heart insufficiency induced by β-adrenoreceptor hyperstimulation with ISO. The ultrastructural features of mitochondria from rat cardiomyocytes were characterized, and the level of oxidative damage, the state of antioxidant systems, and the efficiency of ATP production in cardiac mitochondria of rats with experimental pathology and after a course of treatment with aqTAX were determined.

## 2. Results

### 2.1. Animal Model and Study Design

In this work, a rat model of catecholamine-related myocardial injury was reproduced via a double (with an interval of 24 h) injection of ISO (150 mg/kg body weight), and the possible action of aqTAX (15 mg/kg body weight) was examined after two weeks of its peroral administration. The experimental model recapitulates the main morphofunctional changes in the myocardium in heart failure [4,9,11,25].

Figure 1 and Table 1 demonstrate the experimental design and the main somatic indices of rats at the end of the experiment. Four groups of experimental animals were used: control (Cntl), control + water-soluble taxifolin (Cntl + aqTAX), isoprenaline hydrochloride (ISO), and isoprenaline + aqTAX (ISO + aqTAX) (*n* = 10–11). The doses of the compounds were chosen based on the results of our earlier studies and the literature data [24,26,27,28]. It should be noted that the injection of ISO at a chosen mode led to the appearance of necrotic foci in the left ventricular wall, as evidenced by experiments on cardiac perfusion of the rat heart with Evans blue dye (Appendix A).

It should also be pointed out that ISO caused 33% mortality, with rats dying within the first three days of the experiment. The application of aqTAX reduced the incidence of sudden death in experimental animals with ISO-induced myocardial injury. In addition, an increase (by 24%) in the relative weight of the heart of the animals was recorded in the ISO group, while in the ISO+aqTAX group, this indicator tended to decrease (Table 1).

### 2.2. Electrophysiological Characteristics of the Heart of Experimental Animals

To assess the changes in the cardiac conduction system, an electrocardiogram test (ECG) in experimental animals was performed. Figure 2A demonstrates representative ECG records in four groups of rats. An analysis of ECG data indicated that the injection of ISO caused a decline in the heart rate (HR) and the corresponding elongation of the R–R interval (Figure 2B). This agrees with the literature data that excessive doses of the β-adrenoceptor agonist cause a violation of the sinus rhythm followed by bradycardia at remote terms of observation, indicating the total impairment of contractility and relaxation of the myocardium and the inhibition of the pacemaker function [29,30,31]. In addition, the injection of ISO significantly elongated the QT interval, indicating an extension of the depolarization and repolarization phases relative to the duration of the cardiac cycle. These changes may point to disturbances of the cardiac electrical activity recorded in channelopathies, hypokalemia, and the development of arrhythmias, ischemia, and myocardial infarction, as well as in the long QT syndrome [32].

As seen from the figure, treatment of rats with myocardial injury with aqTAX significantly decreased the QT interval, as compared with the experimental ISO group. The administration of the drug to healthy rats did not affect these parameters, and in the Cntl + aqTAX group, they remained within normal limits. Thus, the administration of aqTAX partially normalized changes in ECG parameters in rats with catecholamine-induced myocardial injury.

### 2.3. Ultrastructural Changes in the Left Ventricular Myocardium and Mitochondria of Experimental Rats

Among all the pathogenetic factors that contribute to the progression of cardiac pathology, mitochondrial alterations are considered the most decisive. Therefore, at the next stage of this study, tissue samples from the anterior left ventricular wall of rats from four experimental groups were examined ultrastructurally using transmission electron microscopy. In the control group, the mitochondria in rat heart cardiomyocytes had a rounded shape, an ordered arrangement of cristae, and a moderately dense matrix (Figure 3A). Mitochondria were located in a strictly ordered manner between myofibrils. AqTAX administered to control animals did not significantly affect the arrangement and the structure of myofibrils and mitochondria in the area of the left ventricular myocardium (Cntl + aqTAX group, Figure 3B).

The injection of ISO to rats promoted degenerative changes in cardiac muscle fibers and mitochondria (Figure 3C). In left ventricular cardiomyocytes, a violation of the ordered arrangement of mitochondria and their association with each other, ruptures of the outer membrane, and the destruction of the cristae membranes were observed. Some of the mitochondria were swollen and had an irregular shape. In organelles, local lysis of cristae and the appearance of vacuoles were seen (Figure 3C).

In the ISO + aqTAX group, the structure and arrangement of myofibrils and mitochondria in the cardiomyocytes of the left ventricle were largely preserved (Figure 3D). Mitochondrial cristae retained their structure, although partially swollen mitochondria were encountered. In the mitochondria of this group, no vacuoles were encountered, and only minor disorders in the orientation and packing of cristae were observed.

A quantitative ultrastructural analysis was carried out to examine the changes in mitochondrial number and morphology in the experimental groups (Figure 4).

One can see that the injection of ISO led to a decrease in the number of the organelles in the left ventricular myocardium (ISO group). In addition, the relative frequency of small mitochondria (up to 2.0 μm) in the ISO group increased. Moreover, the percent of damaged (vacuolated) mitochondria was significantly elevated. A 14-day course of treatment with aqTAX (ISO + aqTAX group) prevented the decrease in the average number and size of mitochondria and also significantly preserved the structure of mitochondrial cristae.

### 2.4. Mitochondrial Respiration and Oxidative Phosphorylation in the Studied Animal Groups

One of the main criteria of the functional activity of mitochondria is the rate of oxidative phosphorylation, which is traditionally measured using high-resolution respirometry (Oxygraph-2k). It follows from the literature data that mitochondrial respiratory complex I (NADH:ubiquinone oxidoreductase) is the least resistant to damage in heart pathologies [12,13,14,15,16]; therefore, we further studied the indicators of respiration of rat heart mitochondria in three metabolic states using the NAD-linked substrates potassium malate and glutamate (Table 2).

It was found that the respiration rate of mitochondria in the phosphorylating (V_3_) and uncoupled states (V_3UDNP_) decreased during the progression of ISO-induced heart damage compared with the control. In addition, a decrease in the respiration control ratio (RCR, V_3_/V_4_) and the ADP/O index and an increase in the time of ADP phosphorylation (T_ph_) were observed compared to the control values. Thus, the development of ISO-induced cardiomyopathy is accompanied by a decline in ATP synthesis in mitochondria and, probably, a disruption of the respiratory complex I.

In the ISO + aqTAX group, an increase in the ADP/O and the RCR parameters and a decrease in the phosphorylation time were noted as compared to the ISO group. Thus, a course of treatment with aqTAX enhances the efficiency of oxidative phosphorylation in cardiac mitochondria from rats with heart injury.

### 2.5. Level of Respiratory Chain Protein Complexes in Rat Heart Mitochondria from Experimental Groups

The changes in respiration and oxidative phosphorylation found in heart mitochondria of rats with ISO-induced heart injury may be associated with decreased expression of subunits of the respiratory complexes. The determination of the protein content of the subunits of five mitochondrial respiratory complexes (NDUFB8, SDHB, UQCRC2, MTCO1, and ATP5A) via the Western blot assay revealed a decrease in the subunits of respiratory complexes I–V in the ISO group relative to the control (Figure 5). It is remarkable that the greatest decrease (by 30%) was recorded in the content of the protein vATP5A, one subunit of ATP synthase (complex V).

One can see that the application of aqTAX (group ISO + aqTAX) led to a statistically significant restoration of the protein level of the mitochondrial respiratory complexes I, II, and V, compared with the ISO group.

### 2.6. Effect of aqTAX Treatment on Oxidative Stress Makers of the Blood Plasma and Heart Mitochondria of ISO-Injured Rats

The accumulation of ROS in cells in pathologies is known to lead to the injury of essential cellular components, including membrane lipids. Figure 6 shows the content of lipid peroxidation products (mainly, the arachidonate by-product malondialdehyde) in four experimental groups, determined via the 2-thiobarbituric acid (TBA) method. It was found that the concentration of TBA-reactive substances (TBRASs) in the blood serum and mitochondria of the heart of rats with cardiomyopathy increased the average twofold. A 14-day course of treatment with aqTAX led to a significant decrease in the level of TBA-active products in both the blood serum and mitochondria, which confirms the antioxidant effect of aqTAX.

As known, an imbalance between the generation and utilization of ROS leads to the development of oxidative stress in mitochondria and the entire cell. Since ISO was reported to increase ROS and exhaust intracellular antioxidants [9,10,11,17], we examined if aqTAX can reverse such oxidative effects of the drug in cardiac mitochondria. Therefore, we determined the activities of the main mitochondrial enzymes of the antioxidant defense system: catalase (CAT), manganese-dependent superoxide dismutase (Mn-SOD), and glutathione peroxidase (GPx) in isolated heart mitochondria from rats of four experimental groups (Figure 6). It was found that the activity of the mitochondrial SOD isoform (Mn-SOD or SOD2) in the ISO group was 34% lower compared to the control. Treatment with aqTAX restored the Mn-SOD activity to the control values. The activity of GPx in heart mitochondria in the ISO group tended to decrease, and the application of aqTAX caused a significant rise in this parameter. The activity of CAT in the control and ISO groups did not significantly differ; however, aqTAX treatment enhanced the activity of this enzyme in animals with ISO-induced cardiomyopathy.

## 3. Discussion

Mitochondrial damage and associated oxidative stress are considered to be major contributory factors in cardiac pathology. In this connection, there is at present a growing understanding that the methods of therapy aimed at regulating the mitochondrial ROS are promising strategies for the correction of heart diseases [14,16,18,19]. One of the effective methods of therapy under the conditions of ROS hyperproduction is considered to be the use of naturally occurring antioxidants, in particular, bioflavonoids, which belong to the P-vitamin group. One of the most potent antioxidants is dihydroquercetin, especially in its water-soluble form. Earlier, we found that aqTAX has significantly more pronounced antihypoxic properties compared to the classical form of TAX. In the aqueous solution of aqTAX, PVP acts as a solubilizer that increases the solubility of TAX. Comparative analysis of ^1^H-NMR spectra of these molecules revealed the formation of complexes of aqTAX with PVP through electrostatic interactions, resulting in its higher water solubility and bioavailability [24].

The current study shows for the first time that aqTAX produces a pronounced protective action in a rat model of ISO-induced myocardial injury, which reproduces key pathological features of sympathetic overstimulation-mediated oxidative damage to the heart in humans. It was demonstrated that the peroral application of the flavonoid aqTAX for 14 days attenuated ultrastructural abnormalities in cardiac mitochondria, including a decrease in the size and the number of these organelles per unit area, and improved the efficiency of oxidation phosphorylation in mitochondria isolated from the cardiac muscle tissue of ISO-treated animals. These beneficial effects of aqTAX are associated with the inhibition of lipoperoxidation parameters and the normalization of the enzymatic activity of GPx and Mn-SOD in rat heart mitochondria. Consistent with the protective action against mitochondrial dysfunction, aqTAX treatment also restored the disturbed intraventricular repolarization as assessed via QT interval duration and improved the blood plasma level of the oxidative stress biomarker TBRASs in animals with ISO-provoked cardiomyopathy.

It has been convincingly demonstrated that high doses of ISO can overstimulate β-adrenergic receptors, primarily the β2 subtype, to provoke oxidative stress, resulting in heart tissue damage [9,10,11]. Our data showed that the introduction of high-dose ISO leads to a 35% mortality in animals. It also increases the relative weight of the heart of rats, probably due to the development of left ventricular hypertrophy, as reported previously [25]. The latter is known to cause the impairment of the contractility function of the myocardium and cardiac failure. As expected, these changes were accompanied by significant disturbances of the atrioventricular conductance, which persisted over a period of two weeks after the administration of ISO. In the ISO group, we noted a decline in the heart rate, a prolongation of the RR interval, and a shortening of the QT interval, which confirms the development of bradycardia and repolarization inhomogeneity in rats with experimental cardiomyopathy. A 14-day course of aqTAX normalized the QT interval duration, indicating a partial restoration of the intraventricular impulse conductance. These findings agree with the literature data indicating that high doses of ISO induce impairments in conductance and hypertrophic changes in the rat myocardial tissue [25,29,30].

At the mitochondrial level, the progression of cardiomyopathy was accompanied by a decrease in the rate of ADP-stimulated and uncoupled respiration of the organelles in the presence of NADH-linked substrates. In addition, we observed a decline in the RCR index and in the ADP/O ratio, as well as an elongation of the phosphorylation time as compared with the control. Hence, the development of cardiomyopathy is related to the decline in oxidative phosphorylation efficiency and the disturbances in the ATP synthesis and, probably, the functioning of the NADH:ubiquinone oxidoreductase (complex I) of the respiratory chain. This is consistent with the literature data indicating that it is this respiration complex that is the least resistant to damage in pathologies accompanied by oxidative stress [12]. A course of treatment with aqTAX led to a significant increase in the RC index and the ADP/O ratio as well as to a decline in the phosphorylation time, as compared with these parameters in rats with cardiomyopathy. Thus, aqTAX is able to restore oxidative phosphorylation in rat heart mitochondria.

The latter may be due to the preservation of the structure and the number of mitochondria in cardiomyocytes. The results of the electron microscope examination, obtained in this study, are consistent with the literature indicating that the administration of high doses of ISO leads to destructive changes in the myocardium [25]. Thus, in the ISO group, a significant rupture and thinning of myofibrils in cardiomyocytes of this area of the heart was observed. Most mitochondria in cardiomyocytes had defects in the arrangement and packing of cristae, an increased space between cristae, and a swollen matrix; according to these features, they were assigned to structurally damaged organelles. In the area of the left ventricle wall, damaged mitochondria, including vacuolated mitochondria with fully and partially destroyed cristae, were seen. In parallel, the number of damaged mitochondria increased, whereas the average number of mitochondria per unit area in this group markedly decreased. Treatment with aqTAX partially restored the structure of mitochondria, in particular cristae membranes, and significantly decreased the number of structurally damaged mitochondria in rat cardiomyocytes. According to the literature data, classical TAX can produce the protective action against common defects in tissue structure through blocking the signaling pathways involved in the development of fibrosis [33,34,35,36].

Our study showed that ISO-induced cardiomyopathy leads to a decrease in the protein level of mitochondrial complexes I, II, and V of the respiratory chain, and the use of aqTAX prevents this decrease. The significant decline in the protein content of ATP synthase (respiratory chain complex V) during the development of cardiomyopathy observed in this study may be due to the existence of a strictly regulated relationship between the level of ATP synthase, ROS formation, and cell death [37,38]. It is known that the defects of the mitochondrial respiratory chain, in particular of ATP synthase, result in mitochondrial dysfunction, which is a crucial cause of heart diseases [39,40].

In heart mitochondria, aqTAX also normalizes the level of lipoperoxidation products in ISO-treated rats. It should be noted that the effect of aqTAX on redox metabolism makes itself evident also at the level of blood serum. It was found that the content of TBA-reactive products in the serum of animals with cardiomyopathy increased more than twofold, and the treatment with aqTAX almost completely normalized this parameter, thereby decreasing the oxidation stress in the animal. Most likely, this effect of the drug is associated with its potent antioxidant and antihypoxic activity [24], and, therefore, it effectively prevents the oxidative damage to mitochondria. According to the literature data, classical TAX also provides the defense of membrane lipids against peroxidation in tissues [36].

A direct correlation between the TBRAS levels in the serum and in the mitochondria isolated from metabolically expensive tissues of experimental animals has been also observed earlier [26,31]. Hence, it may be suggested that the TBRAS level measured in the blood serum may be of prognostic significance and indicate the progression of oxidative stress and associated mitochondrial damage in the heart tissue of a patient.

The present study also showed that elevated TBRAS levels are associated with decreased activity of key antioxidant defense systems in the heart mitochondria of rats with ISO-induced cardiomyopathy. This may indicate that the antioxidant defense systems are unable to detoxify the increased amounts of ROS in the mitochondria; as a result, they can accumulate and promote the development of oxidative stress in cardiomyocytes. A course of treatment with aqTAX led to a significant increase in the enzymatic activities of the mitochondrial superoxide scavenging enzyme MnSOD and glutathione peroxidase, whose preferred substrate is hydrogen peroxide. It should be noted that aqTAX is a much stronger antioxidant than classical TAX and some other antioxidants. In a primary neural cell culture model, aqTAX affects a greater number of protective genes than TAX and more effectively enhances the basal and the oxygen-glucose deprivation//reoxygenation (OGD/R)-induced expression of genes encoding ROS scavenging, anti-inflammatory, and antiapoptotic proteins [24]. Altogether, this permits us to consider aqTAX as a potential cardioprotector in the complex therapy of pathologies, including cardiovascular complications of COVID-19, as previously proposed [41].

This study has some limitations. In particular, when reproducing the experimental model, animals were selected randomly, regardless of their individual resistance to stress conditions and hypoxia. Further studies are needed to understand the complex interaction between the morphofunctional changes in the mitochondrial apparatus, stress resistance, and serum catecholamine levels, as well as to identify risk factors and develop personalized strategies for the prevention and treatment of stress-induced cardiomyopathy, which is common in patients with pheochromocytoma and in the presence of elevated levels of endogenous and/or exogenous catecholamines.

## 4. Materials and Methods

### 4.1. Rat Model of Isoprenaline-Induced Myocardial Injury

All experimental protocols were approved by the Commission on Biosafety and Bioethics (Institute of Theoretical and Experimental Biophysics, Russian Academy of Sciences; Permission no. 16 of 18 March 2024). Rats received humane care in accordance with the Principles of Laboratory Animal Care declared by the Directive 2010/63/EC of the European Parliament. All experiments were carried out in compliance with the Rules for Conducting Research with Experimental Animals (Order of the Ministry of Health of Russia, 12 August 1997, no. 755). Adult Wistar male rats (280–310 g) were maintained in a temperature-controlled room on a 12/12 h dark/light cycle. Before the start of the experiments, animals were given a two-week adaptation period. The damage to the myocardium in rats was initiated via a subcutaneous (SQ) injection of the agonist of β-adrenergic receptors isoprenaline hydrochloride (ISO) freshly dissolved in normal saline (NS) according to the conventional scheme [25,27,31].

The animals were divided into four experimental groups of ten animals each (Table 3). Control animals received subcutaneous injections of saline (100 µL/100 g body weight), which were made twice with an interval of 24 h, and normal drinking water. The animals of the second group (Cntl + aqTAX), after being injected with NS, were daily watered with aqTAX at a dose of 15 mg/kg body weight for 14 days. Animals placed in individual cages received 10 mL of a freshly prepared mixture of aqTAX and drinking water at night for two weeks. The ISO group included animals that were injected subcutaneously twice with an interval of 24 h with ISO at a dose of 150 mg/kg body weight. The rats in the ISO group received normal drinking water. The animals of the ISO + aqTAX group received two injections of ISO and were given for the next 14 days aqTAX at a dose of 15 mg/kg body weight. The doses were chosen with consideration of a high intensity of metabolism in rats and correspond to those used in other studies [24,26,27,28].

All animals were taken out of the experiment 15 days after the first injection of ISO or physiological saline, respectively. For further visualization of damage to myocardial tissue, foci of necrosis were detected via perfusion of the heart with Evans blue dye according to the standard technique [42]. In this series of experiments, before euthanasia, rats were administered 1% Evans blue dye at a dose of 2 mL/kg. After 30 min, the hearts of the rats were immediately removed and photographed.

### 4.2. Electrocardiography and Analysis of Electrocardiograms

Electrocardiograms (ECG) were recorded in standard lead II using a SparkFun heart rate monitor (SparkFun Electronics, Niwot, CO, USA) [43]. The immobilization of animals was performed via combined anesthesia using Zoletil 100 (VirbacSanteAnimale, Carros Cedex, France) at a dose of 50 mg/kg body weight, intramuscularly, with premedication with Rometar (AO Bioveta, Moscow, Russia) as a myorelaxant at a dose of 10 mg/kg, intramuscularly. Narcosis was verified from the disappearance of response to painful stimuli (prick of the paw) and the suppression of the corneal reflex in animals, after which they were placed in the chamber. ECG signals were recorded for 10 min via a computer instrumented with the Clampfit 11.2.0.59 Software (Molecular Devices LLC, San Jose, CA, USA). The analysis was performed using the data of continuous recording. ECG signals were filtered using a high-pass Bessel filter and a low-pass Gaussian filter. The heart rate (as beats per minute, bpm) and other most significant ECG parameters were analyzed according to the latest recommendations [44,45,46]. After decapitation of an animal, the heart was weighed, and the samples of the tissue were taken for further examination.

### 4.3. Transmission Electron Microscopy

Ultrastructural changes in cardiomyocytes were identified via transmission electron microscopy examination of tissue samples from the anterior wall of the left ventricle of experimental rats using a JEM-100B electron microscope (JEOL, Tokyo, Japan). Samples of the tissue were fixed for 2 h with a 1% solution of osmium acid in PBS, dehydrated with alcohols of increasing concentration, and incapsulated in Epon 812 resin. Ultrathin sections (70–75 nm) were cut with a Leica EM UC6 microtome (Leica Microsystems, Wetzlar, Germany) and stained with uranyl acetate and lead citrate.

The morphometric analysis of tissue areas on the microphotographs was performed using Image J2 (Fiji) software (National Institutes of Health, Bethesda, MD, USA). The ultrastructure of myocardial tissue, including mitochondrial morphology, relative distributions of mitochondrial size (perimeter), and the average number of the organelles, were observed. To quantify the number and perimeter of mitochondria, we analyzed 30 different fields in each experimental group (*n* = 3). Stretched mitochondria with increased intermembrane space and misshapen cristae lumens were quantified as swollen. Intact organelles were defined as mitochondria with continuous inner and outer membranes, a compact intermembrane space, and orderly cristae that were folded into a compact matrix.

### 4.4. Isolation of Rat Heart Mitochondria

The heart tissue was placed rapidly in an isolation medium cooled to 0 °C, which contained 210 mM mannitol, 70 mM sucrose, 10 mM HEPES (pH 7.4), 0.5 mM EGTA, 1 mM EDTA, and 0.5% defatted bovine serum albumin (BSA). The heart was cut into four parts, washed with physiological saline three times, laid onto filter paper, and dried. The isolation of rat heart mitochondria was carried out at 4 °C as described [31,47]. The tissue was homogenized with a T 25 digital Ultra Turrax (IKA-Werke GmbH and Co. KG, Staufen, Germany) disperser. The centrifugation was performed at 1200× *g* for 3 min and at 1800× *g* also for 3 min. The supernatant was taken and centrifuged at 12,000× *g* for 10 min. The mitochondrial pellet was placed in a BSA-free medium containing 210 mM mannitol, 70 mM sucrose, 10 mM HEPES (pH 7.4), and 0.1 mM EGTA in a ratio of 1:10. The pellet was gently resuspended three to four times using a glass homogenizer with the addition of the medium (0.1 mL of medium per 1 g of the heart tissue) and centrifuged again.

The final mitochondrial pellet was homogenized manually with a Teflon pestle and poured into an Eppendorf tube. The mitochondrial protein concentration was measured via the Lowry method on a Shimadzu UV-2401PC spectrophotometer (Kyoto, Japan) at a wavelength of 750 nm using the calibration curve obtained with the use of defatted BSA.

### 4.5. High-Resolution Respirometry

The respiration of rat heart mitochondria was defined with high-resolution O2K respirometry in an incubation medium containing 100 mM KCl, 75 mM mannitol, 25 mM sucrose, 5 mM KH_2_PO_4_, 10 mM HEPES/KOH (pH 7.4), and 0.5 mM EGTA. As respiration substrates, 5 mM potassium glutamate and 5 mM potassium malate were used. The respiration rates of mitochondria in different metabolic states were studied using a high-resolution Oroboros Oxygraph-2k respirometer with the DatLab 4 software (Oroboros Instruments GmbH, Innsbruck, AT, Austria).

The functional state of mitochondria (0.5–0.6 mg/mL) was analyzed by calculating the rate of oxygen uptake in the presence of 200 μM ADP (V_3_) and after its phosphorylation into ATP (V_4_), as well as the rate of uncoupled respiration in the presence of 50 μM 2,4-dinitrophenol (V_DNP_). Mitochondrial oxidative phosphorylation was assessed from the V_3_/V_4_ ratio and the respiratory control (RC), the ADP/O ratio (the amount of added ADP to the amount of oxygen consumed), and the time spent for the phosphorylation of added 200 µM ADP [48].

### 4.6. Electrophoresis and Immunoblotting

The samples of cardiac mitochondria (27–35 mg/mL) were dissolved in Laemmli buffer (Bio-Rad, Hercules, CA, USA) and run on 12.5% SDS–PAGE at a loading amount of 10 μg per each lane. The mitochondrial proteins were transferred from the gel onto a 0.45 µm nitrocellulose membrane (Cytiva, Marlborough, MA, USA). After overnight blocking with 2% non-fat milk buffer, the membranes were stained with the appropriate primary antibody. The total OXPHOS Rodent WB Antibody Cocktail (ab110413) and anti-VDAC1 (ab15895) used to normalize the protein level were from Abcam (Abcam, Cambridge, UK). A rat heart tissue lysate–mitochondrial extract (ab110341) (Abcam, Cambridge, UK) was applied as a positive control (PC). Protein bands were determined by using goat anti-rabbit IgG antibody conjugated to horseradish peroxidase (cat.No. 7074, Cell Signaling technology Inc., Danvers, MA, USA) for chemiluminescent detection. Horseradish peroxidase activity was monitored with ECL reagents (Bio-Rad, Hercules, CA, USA) using a LI-COR system (LI-COR, Lincoln, NE, USA) with the LI-COR Image Studio software version 5.2.

### 4.7. Determination of the Content of Lipoperoxidation Products

The level of thiobarbituric acid-reactive substances (TBRASs) in the blood plasma and heart mitochondria from the experimental groups was determined as described [49]. The optical density was measured on a Shimadzu UV-2401PC spectrophotometer (Kyoto, Japan) at wavelengths of 532 and 650 nm vs. blank samples.

### 4.8. Assay of the Enzymatic Activity of Antioxidant Defense Systems in Rat Heart Mitochondria

The activity of manganese superoxide dismutase, glutathione peroxidase, and catalase in isolated mitochondria was determined as described earlier [50]. Rat heart mitochondria were osmotically disrupted in 5 mM potassium phosphate buffer (pH 7.4) and subjected to three freezing/thawing cycles. Then, the probes were centrifuged at 12,000× *g* for 10 min. The sediment was exposed to a lysis mixture (0.5% RIPA buffer with protease inhibitor cocktail), and the activity of membrane-bound enzymes was analyzed in the lysate with the help of a Shimadzu UV-2401PC spectrophotometer (Kyoto, Japan).

The enzymatic activity of manganese superoxide dismutase (Mn-SOD) was determined by measuring the rate of inhibition of nitro blue tetrazolium chloride (NBT) reduction in the xanthine–xanthine oxidase system [51]. In these experiments, Mn-SOD activity was defined in the presence of the Cu/Zn-SOD inhibitor potassium cyanide (1mM) in a medium containing 50 mM potassium phosphate (pH 10.2), 0.1 mM EDTA, 62.5 μM NTB, 0.5 mM xanthine, and 0.03 U/mL XO.

The glutathione peroxidase activity was evaluated by measuring the decrease in absorbance at 340 nm due to the NADPH oxidation in the presence of H_2_O_2_ and GSH. The reaction medium consisted of a buffer containing 50 mM phosphate buffer (pH 7.0), 1 mM KN_3_, 1 mM Na-EDTA, 5 mM GSH, 0.4 U/mL glutathione reductase, 200 μM NADPH, and 180 μM H_2_O_2_.

The activity of catalase in isolated mitochondria was assessed via light absorption at 240 nm using H_2_O_2_ as a substrate [52]. The reaction medium contained H_2_O_2_ at a final concentration of 30 mmol/L.

### 4.9. Statistical Data Processing

The data obtained were processed using the GraphPadPrism 8.0 and Excel 6.0 software and presented as the mean ± standard error of the mean. The analysis of the normality of data distribution was performed using the Shapiro–Wilk test. If the data were within the normal distribution, the statistical processing of the results was carried out using a one-way ANOVA. Further comparison of the means of the dispersion complex was carried out using the Tukey post hoc test. The significance level was taken to be *p* < 0.05.

## 5. Conclusions

To summarize, we have demonstrated in this study that aqTAX prevents the disturbance of oxidative phosphorylation and restores redox metabolism as well as the ultrastructure of cardiac mitochondria of rats with ISO-induced myocardial injury. Consistent with the protective action against mitochondrial dysfunction, aqTAX treatment partially restored the disturbed ECG profile and improved the blood plasma level of the oxidative stress biomarker TBRAS in animals with ISO-provoked cardiomyopathy. Presumably, the positive effects of aqTAX are due to its potent antioxidant properties and enhanced water solubility, resulting in the suppression of oxidative damage to mitochondria in the myocardium. These effects can also be associated with the ability of this compound to effectively enhance basal and hypoxia-induced expression of genes encoding ROS-detoxifying systems, anti-inflammatory, and anti-apoptotic proteins. Taken together, these data allow one to consider this compound as a potential cardioprotector in the complex therapy of heart failure.

## Figures and Tables

**Figure 1 ijms-25-11596-f001:**
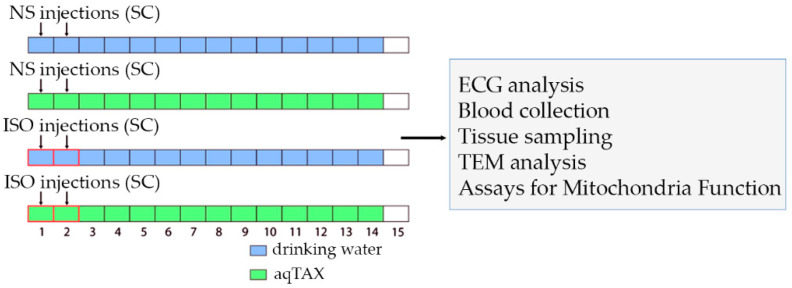
Experimental study design. Male Wistar rats were randomly assigned to the following groups: the control (Cntl) group was given two subcutaneous (SC) injections of normal saline (NS); the control + aqTAX (Cntl + aqTAX) group was watered with aqTAX (aqTAX, 150 mg/kg/day body weight, *per os*); the isoprenaline (ISO) group was treated with isoprenaline hydrochloride (ISO, 150 mg/kg body weight, s.c.); the isoprenaline + aqTAX (ISO + aqTAX) group received two injections of ISO and were given a freshly prepared mixture of aqTAX and drinking water for the next 14 days. Rats (*n* = 10 *per* group) were sacrificed 15 days after the first injection, and the above analyses were carried out.

**Figure 2 ijms-25-11596-f002:**
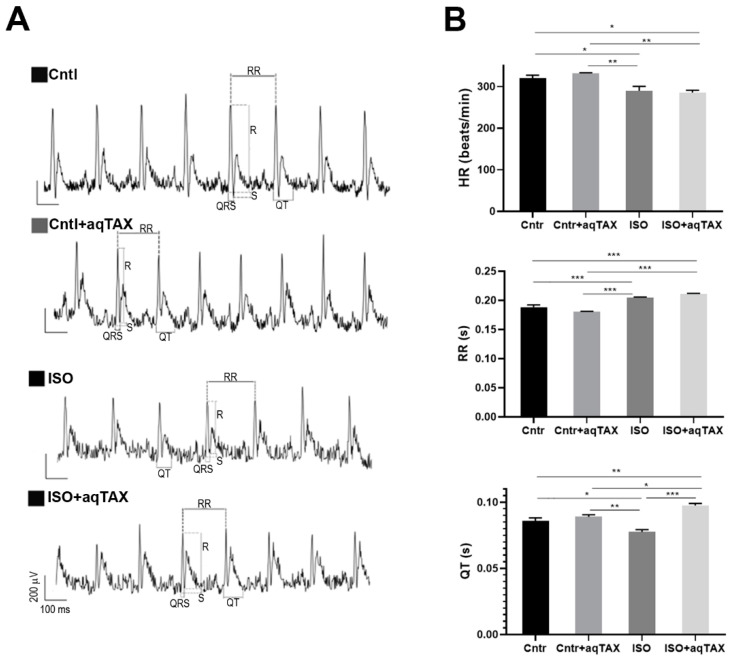
Electrophysiological characteristics of the heart of experimental animals. (**A**) Representative lead II ECG records. (**B**) Heart rate (beats per minute) and the most common ECG intervals: RR and QT (in seconds) in the studied animal groups (control (Cntl), the control + aqTAX (Cntl + aqTAX) group, the isoprenaline hydrochloride (ISO) group, and the isoprenaline hydrochloride + aqTAX (ISO + aqTAX) group. Data are presented as mean values ± SEM from a group of five rats. Statistical significance was analyzed using one-way analysis of variance and Tukey’s post hoc test. * *p* < 0.05; ** *p* < 0.01; *** *p* < 0.001.

**Figure 3 ijms-25-11596-f003:**
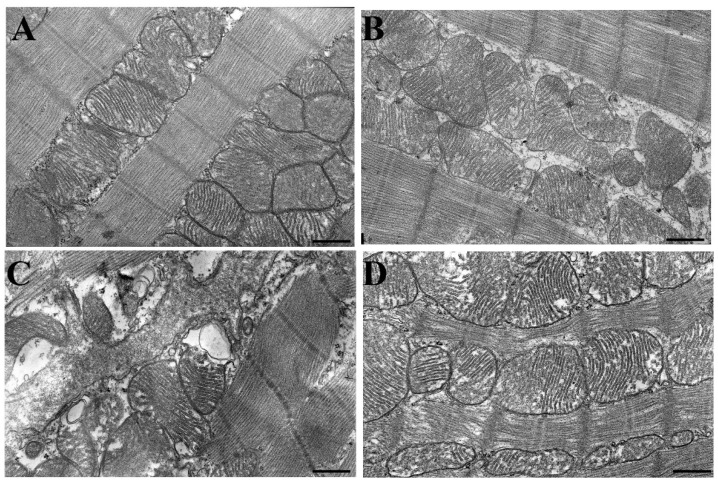
Electron micrographs of mitochondria in the subsarcolemmal region of a rat cardiomyocyte from the left ventricle of the heart: (**A**) the control group (Cntl); (**B**) control rats after a course of treatment with aqTAX (Cntl + aqTAX); (**C**) isoprenaline-induced myocardial injury (ISO); (**D**) ISO-injured rats after a course of treatment with aqTAX (ISO + aqTAX). Representative micrographs from three independent experiments are shown. Lead citrate and uranyl acetate × 5000. Scale bars, 1 µm.

**Figure 4 ijms-25-11596-f004:**
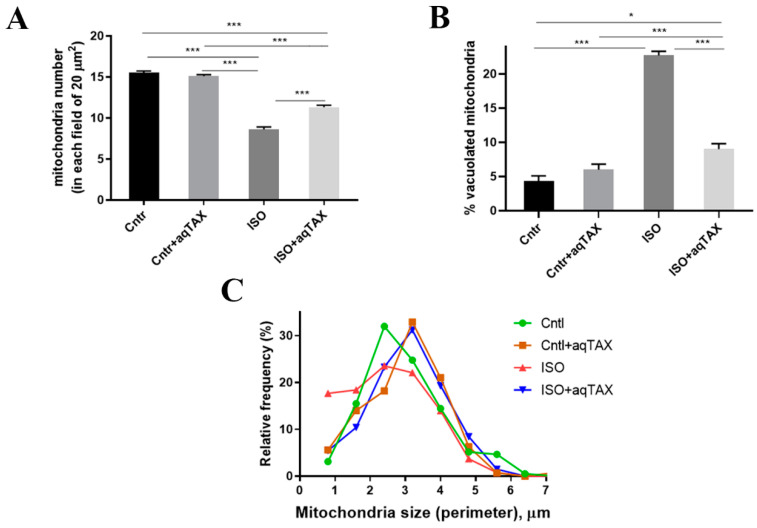
Morphometric analysis of rat heart subsarcolemmal mitochondria in the studied animal groups (control (Cntl), control + aqTAX (Cntl + aqTAX), isoprenaline hydrochloride (ISO), and isoprenaline hydrochloride + aqTAX (ISO + aqTAX)). (**A**) Average number of the organelles in rat cardiomyocytes, normalized to the area (mm^2^) of the TEM images analyzed. (**B**) Bar graph summarizing the number of damaged (vacuolated) mitochondria (% of the control) between conditions. (**C**) Relative distribution of mitochondrial size (perimeter, µm) in rat cardiomyocytes under different treatments. The assays were carried out from at least three rat left ventricles in each group. At least 30 fields of view per group were analyzed. Data are presented as mean values ± standard derivations. * *p* < 0.05; *** *p* < 0.001.

**Figure 5 ijms-25-11596-f005:**
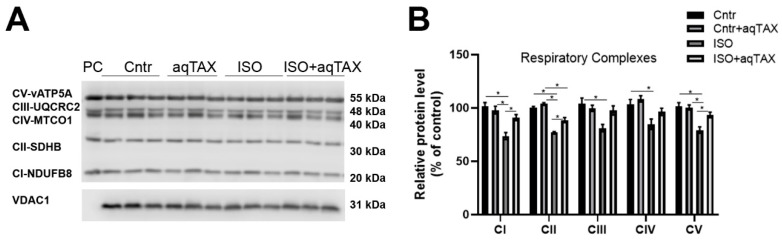
Content of the subunits of five respiratory chain complexes (CI-NDUFB8, CII-SDHB, CIII-UQCRC2, CIV-MTCO1, and CV-vATP5A) in rat heart mitochondria from the experimental groups. (**A**) Representative Western blot of typical oxidative phosphorylation (OXPHOS) subunits of complexes I–V in the mitochondria isolated from the organs of experimental animals. The analysis was performed using the Total OXPHOS Rodent WB Antibody Cocktail and anti-VDAC1 antibodies (Abcam, #ab110413 and #ab15895, Cambridge, UK; 10 μg of mitochondrial protein per lane). PC–positive control (Abcam, #ab110341, Cambridge, UK). (**B**) Densitometric analysis of the level of the OXPHOS subunits in the cardiac mitochondria of experimental animals. Data are expressed as mean values ± standard errors. Control values are taken as 100%. Significant differences tested with a one-way analysis of variance and Tukey’s post hoc test. * *p* < 0.05.

**Figure 6 ijms-25-11596-f006:**
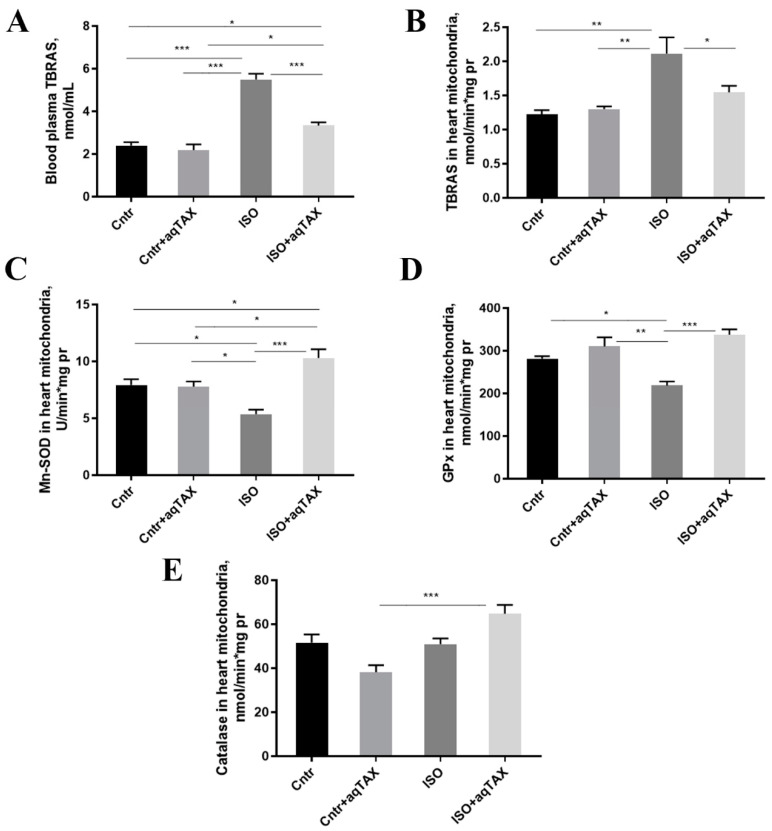
Oxidative stress-related makers of the blood plasma and heart mitochondria of rats from the experimental groups: control (Cntl), control + aqTAX (Cntl + aqTAX), isoprenaline hydrochloride (ISO), and isoprenaline hydrochloride + aqTAX (ISO + aqTAX). (**A**) Plasma levels of thiobarbituric acid reactive substances (TBARS). (**B**) TBARS levels in isolated rat heart mitochondria. (**C**) Enzymatic activities of the superoxide scavenging enzyme manganese superoxide dismutase (MnSOD) in rat heart mitochondria. (**D**) Glutathione peroxidase (GPx) enzyme activities in rat heart mitochondria. (**E**) Enzymatic activities of catalase in rat heart mitochondria. Values are mean ± SEM from five animals in each group. Significant differences tested with a one-way analysis of variance and Tukey’s post hoc test. * *p* < 0.05; ** *p* < 0.01; *** *p* < 0.001.

**Table 1 ijms-25-11596-t001:** Main somatic indices of animals at the end of the observations.

Group	Body Weight (BW) (g)	Heart Weight (HW) (g)	HW/BW Ratio (mg/g)	Mortality Rate (%) (Dead/Alive)
Cntl	316.33 ± 17.45	1.19 ± 0.09	3.75 ± 0.13	0 (0/10)
Cntl + aqTAX	314.60 ± 12.07	1.18 ± 0.03	3.88 ± 0.14	0 (0/10)
ISO	297.00 ± 7.28	1.37 ± 0.03	4.64 ± 0.15 *	33.3 (5/15)
ISO + aqTAX	281.00 ± 8.14	1.23 ± 0.03	4.27 ± 0.13	9 (1/11)

Values are means with standard errors of the mean. One-way analysis of variance followed by Turkey’s post hoc *test* was used (*n* = 10–15). * *p* < 0.05 compared to the Cntl group.

**Table 2 ijms-25-11596-t002:** Mitochondrial respiration and oxidative phosphorylation in the groups studied.

Group	V Respiration, nmol O_2_·min^−1^·mg^−1^ pr			RCR (State 3/State 4)	ADP/O	T_ph,_ s
	State 3	State 4	State 3U_DNP_			
Cntl	69.5 ± 3.2	10.2 ± 1.2	70.7 ± 1.3	8.2 ± 0.4	3.12 ± 0.07	32.8 ± 0.9
Cntl + aqTAX	71.4 ± 2.0	8.5 ± 0.9	69.2 ± 3.0	8.8 ± 0.9	3.22 ± 0.08	27.8 ± 0.7
ISO	52.9 ± 3.3 *	9.3 ± 0.8	58.0 ± 2.5 *	5.2 ± 0.2 *	2.66 ± 0.09 *	39.5 ± 2.4 *
ISO + aqTAX	67.5 ± 2.5 *^,#^	9.6 ± 1.0	71.6 ± 3.0 *^,#^	7.8 ± 0.7 *^,#^	3.25 ± 0.11 *^,#^	29.7 ± 1.4 *^,#^

Respiration of rat heart mitochondria was fueled by 5.0 mM potassium glutamate and 5.0 mM malate. Medium composition: 100 mM KCl, 75 mM mannitol, 25 mM sucrose, 5 mM KH_2_PO4, 10 mM HEPES/KOH (pH 7.4), and 0.5 mM EGTA. Mitochondrial respiration in States 3 and 3U_DNP_ was initiated by ADP (200 µM) and 2,4-dinitrophenol (50 µM), correspondingly. The results are expressed as means ± SEM (*n* = 5–6). Statistical significance was analyzed using one-way analysis of variance and Tukey’s post hoc test. * *p* < 0.05 compared to the Cntl group; ^#^ *p* < 0.05 compared to the ISO group.

**Table 3 ijms-25-11596-t003:** Experimental groups of rats.

Name	SQ Injections		aqTAX, mg/kg	Time After ISO (NS) Exposure
	Normal Saline (NS), mL/kg	Isoprenaline (ISO), mg/kg/day for 2 Days		
Cntl	1.2	-	-	14 days
Cntl + aqTAX	1.2	-	15	14 days
ISO	-	150	-	14 days
ISO + aqTAX	-	150	15	14 days

## Data Availability

The data presented in this study are available upon request from the corresponding author.

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
