# Peer review of "Pathological Alterations in Heart Mitochondria in a Rat Model of Isoprenaline-Induced Myocardial Injury and Their Correction with Water-Soluble Taxifolin"

_ijms, 2024, doi:10.3390/ijms252111596_

Round 1

Reviewer 1 Report

Comments and Suggestions for Authors

The authors offer new, innovative insight into the effect of a 14-day course of oral administration of water-soluble taxifolin (aqTAX, 15 mg/kg) on ​​the progression of ultrastructural and functional abnormalities in mitochondria and electrical activity of the heart in an isoprenaline-model of myocardial injury. In addn., the proposed manuscript seeks to determine the delayed ISO-induced myocardial injury, accompanied by an increase in the duration of RR and QT intervals, and long-term administration of aqTAX partially restores impaired intraventricular conduction. Innovatively, it has been shown that ISO injections lead to profound ultrastructural changes of myofibrils and mitochondria in cardiomyocytes in the left ventricular myocardium, including disruption of the orderly arrangement of mitochondria between myofibrils, as well as a reduction in the size and number of these organelles per unit area. In addition, there was a decrease in the protein level of the subunits of complexes I-V of the respiratory chain and the activity of the antioxidant enzymes glutathione peroxidase, catalase and Mn-SOD in mitochondria. The manuscript, written in standard English, clinically focuses on the administration of aqTAH leading to an increase in the efficiency of oxidative phosphorylation and a partial restoration of the morphometric parameters of mitochondria in cardiac tissue of rats with experimental pathology. In addition, the authors confirm their thesis by noting that the beneficial effects of aqTAX are related to the inhibition of lipid peroxidation and normalization of the enzyme activities of Mn-SOD and glutathione peroxidase in heart mitochondria. The present study sheds new light/in great detail on the potential use of aqTAX as a cardioprotectant in the complex therapy of heart failure.

The abstract is detailed and well written. Adding a graphic abstract is also appropriate. The proposed introduction is well-formed and the stated purpose is systematized. Materials and methods are described in detail and laid out without unnecessary weighting of the text. The figures are readable, well laid out, and comprehensively described. The presentation of results and discussion is original and comprehensive. The proposed conclusion is good, but the complementation with limitations and future research is unclear. To be added. The references used fully confirm the stated thesis.

Reviewer 2 Report

Comments and Suggestions for Authors

Dear Editor

I have gone through the manuscript titled ‘Structural and Functional Alterations in Heart Mitochondria in a Rat Model of Isoprenaline-induced Myocardial Injury and Their Correction with Water-Soluble Taxifolin’. Manuscript is well written and presented but some revision is required to accept the manuscript for the publication.

Introduction: write one paragraph about Taxifolin, sources in which it is abundantly found, the structure of Taxifolin, pharmacokinetics parameters, and water solubility as compared to quercetin.

Why was a water-soluble compound selected for study? Water soluble compounds can easily be excreted in unabsorbed form from the body.

Why was the standard compound not used to treat or cure the same disease conditions, to get better comparison data of results? Without a standard compound, how can authors justify that Taxifolin is a potential cardioprotector in the complex therapy of heart failure?

water-soluble complex of TAX with polyvinylpyrrolidone (PVP), why need to make a complex with PVP? Did the same complex used for the this study and why?

Check the manuscript for typos, sentences, and grammatical errors.

Comments on the Quality of English Language

English needs to improve.
